# Attention to Face as a Predictor of Developmental Change and Treatment Outcome in Young Children with Autism Spectrum Disorder

**DOI:** 10.3390/biomedicines9080942

**Published:** 2021-08-02

**Authors:** Kenza Latrèche, Nada Kojovic, Martina Franchini, Marie Schaer

**Affiliations:** 1Department of Psychiatry, Faculty of Medicine, University of Geneva, 1206 Geneva, Switzerland; nada.kojovic@unige.ch (N.K.); marie.schaer@unige.ch (M.S.); 2Fondation Pôle Autisme, 1204 Geneva, Switzerland; martina.franchini@pole-autisme.ch

**Keywords:** autism spectrum disorders, eye-tracking, social attention, early intervention, predictors of treatment outcome

## Abstract

The beneficial effect of early intervention is well described for children with autism spectrum disorder (ASD). Response to early intervention is, however, highly heterogeneous in affected children, and there is currently only scarce information about predictors of response to intervention. Based on the hypothesis that impaired social orienting hinders the subsequent development of social communication and interactions in children with ASD, we sought to examine whether the level of social orienting modulates treatment outcome in young children with ASD. We used eye-tracking technology to measure social orienting in a group of 111 preschoolers, comprising 95 young children with ASD and 16 children with typical development, as they watched a 29 s video of a woman engaging in child-directed speech. In line with previous studies, we report that attention to face is robustly correlated with autistic symptoms and cognitive and adaptive skills at baseline. We further leverage longitudinal data in a subgroup of 81 children with ASD and show that the level of social orienting at baseline is a significant predictor of developmental gains and treatment outcome. These results pave the way for identifying subgroups of children who show a better response to early and intensive intervention, a first step toward precision medicine for children with autism.

## 1. Introduction

Autism spectrum disorders (ASDs) refer to a group of neurodevelopmental disorders characterized by marked deficits in social communication and social interactions, along with restricted, repetitive behaviors or interests [1]. Among the several early signs of ASD that have been studied [2,3,4], diminished attention toward social information, or impaired social orienting, emerges as the earliest and the most robust [5,6,7]. Social orienting refers to the child’s spontaneous attention toward social information [8]. According to the social motivation theory in ASD, less social orienting in early development leads to decreased social learning experiences in toddler years, which then impacts social cognition development and social skills [9,10,11]. In other words, diminished attention to social information in early years has detrimental cascading effects on the development of social cognition and communication [12,13]. 

Eye-tracking is a powerful technology to measure social orienting in autism. Studies have used stimuli of faces, people, and body motion to compare social orienting in young children with ASD and their typically developing (TD) peers [14]. A majority of eye-tracking studies indeed reported less attention to faces in terms of fixation time in young children with ASD when compared to their TD peers [7,13,15,16,17,18,19]. However, some studies [19,20] do not find differences in attention to faces between children with ASD and their TD peers. These contrasting results in attention to faces may be explained by the presentation of the face stimulus. One study [21] focusing on attention to faces in three conditions (static, dynamic, speaking) found that children with ASD showed less visual attention than their TD peers only when the face was speaking. Another relevant explanation is related to social context, which refers to the social stimuli (i.e., number of people involved) and its complexity (i.e., with or without social interactions, child-directed speech, eye contact) [9,14,19]. Indeed, young children with ASD orient less toward the face than their TD peers when the stimulus is a person engaging in explicit cues for social attention, e.g., eye contact and child-directed speech [7,13,16,22]. By contrast, the difference in attention to faces between young children with ASD and their TD peers is less evident when the social stimulus is of reduced complexity, e.g., the person does not use eye contact or speech to direct attention. In other words, young children with ASD show decreased attention to faces when the stimulus is more socially engaging [21] and as social complexity increases [17]. Consistent with the social motivation theory [9,10,11], less social attention toward complex social stimuli thus compromises social learning experiences, which then impacts the development of social skills emerging in early development (e.g., joint attention) [9].

Eye-tracking measures of social orienting have been associated with the level of autistic symptoms and cognitive skills. Using a silent task featuring social stimuli (children doing yoga) on one side of the screen and geometric stimuli (geometric shapes) on the other side, Pierce et al. [4] showed that toddlers with ASD who had a visual preference for the social side of the screen (i.e., Social Responders, SR) had fewer autistic symptoms and better cognitive skills than the toddlers with ASD who had a visual preference for the geometrical side of the screen (Geometric Responders, GR). From a longitudinal perspective, social orienting is also relevant to identify different patterns of visual exploration in children with ASD, leading to distinct developmental trajectories. Using a visual preference task adapted from the GeoPref Test published by Pierce et al. [5], we previously found that young children with ASD who were SR at baseline showed reduced severity of symptoms a year later [23]. This finding was replicated by another recent study [24], which also found greater cognitive gains for the SR group. Therefore, the level of social orienting at baseline appears to be a promising predictor of early developmental trajectories and outcomes of young children with ASD [23,24]. 

To our knowledge, studies have yet to explore whether social orienting could modulate the responses to treatment. While all early and intensive intervention approaches aim to support social skill and language development and learning for children with ASD, there is high heterogeneity in responses to treatment, with some children progressing rapidly and others showing slower gains [25,26,27]. Since the last decade, the Early Start Denver Model (ESDM), has been recognized as an empirically validated early and intensive intervention [28]. In a randomized control trial [29], Dawson et al. evaluated the efficacy of the ESDM, which is an early and intensive intervention based on social motivation. By comparing a group of young children with ASD who received the ESDM for two years to another group of children who were assessed yearly with intervention recommendations, the authors [29] found that the ESDM group showed increased cognitive skills and adapting functioning, and reduced symptom severity. Considering that the ESDM recalibrates the social motivational system by focusing on communication, cognitive, motor, and adaptive behavior, it is crucial to explore whether children with higher levels of social orienting at baseline show better treatment outcomes. To our knowledge, there is scarce information about predictors of outcome for early and intensive interventions, especially in regard to specific approaches such as the ESDM. 

Given the high heterogeneity in treatment outcome and the absence of empirically validated predictors of response to the ESDM, this study aims to better understand these heterogenous outcomes by investigating the predictive role of a biological biomarker, namely an eye-tracking measure of attention to face. The aims of our study are threefold: (1) from a cross-sectional standpoint, to examine the differences of attention to face between an ASD group and a TD group; (2) to investigate the relationship between attention to face and clinical characteristics in terms of autistic symptoms and developmental and adaptive skills in a sample of 95 young children with ASD; (3) from a longitudinal perspective, to investigate whether levels of attention to face at baseline had an impact on developmental change over time, in a longitudinal subsample of 85 young children with ASD with 265 visits; and (4) to explore whether children’s levels of attention to face at baseline had an impact on treatment outcome, whether the children received an early and intensive intervention (ESDM) or treatment in the community (community treatment, CT). First, based on previous studies [19,20] stating that diminished attention to face in children with ASD was more evident when presented with socially complex scenes, we hypothesized that children with ASD and their TD peers would not demonstrate a different level of attention to face. Second, given that social orienting is essential to learn through social experiences [5,6,9], we hypothesized that more attention to face would be associated with fewer symptoms related to social communication and interaction and better developmental and adaptive skills at baseline in children with ASD. Third, considering the importance of social orienting in the development of social skills [23,24], we hypothesized that more attention to face at baseline would predict higher developmental gains over time. Fourth, given the validated efficacy of early and intensive interventions, such as the ESDM [27], we hypothesized that more attention to face at baseline would predict better treatment outcome for the subgroup receiving the ESDM in comparison to the CT group, and that less attention to face at baseline would have more detrimental consequences for the outcome of the CT subgroup than for the ESDM subgroup. Therefore, such results would allow us to identify subgroups of children with ASD who show a better response to early and intensive intervention, which is a first important step toward precision medicine.

## 2. Materials and Methods

### 2.1. Participants

#### 2.1.1. The Context of the Current Study

The data for the present study were acquired as part of the Geneva Autism Cohort, which is described in previous publications [23,24]. The Geneva Autism Cohort is an ongoing, longitudinal cohort study that started in 2012, that assesses preschool-aged children with ASD and their TD peers. The objective of the ongoing, longitudinal cohort study is to better understand the developmental trajectories of young children with ASD, in relation to their response to treatment. In order to examine their developmental change, participants are assessed every six months for a duration of two years. Assessments include developmental evolution (Mullen Scales of Early Learning, MSEL) [30], adaptive behaviors (Vineland Adaptive Behavior Schedule—second edition, VABS-II) [31], and autistic symptoms (Autism Diagnostic Observation Schedule, second edition, ADOS-2) [32]. This study protocol was approved by the Ethics Committee of the Faculty of Medicine of Geneva University, Switzerland and all parents gave written informed consent for their child to participate. The children with ASD were recruited through local clinical centers specialized in developmental disorders and parent associations. For children with ASD, the diagnosis was confirmed using the Autism Diagnostic Observation Schedule, second edition—ADOS-2 [32]. The TD participants were recruited through announcements in the Geneva community. The TD children were also assessed using the ADOS-2 to ensure the absence of ASD symptoms prior to their inclusion in the TD group. 

The present study is based on an eye-tracking task that we designed in 2018. Our task was specifically designed for children younger than 4, so only a subset of participants from the whole Geneva Autism Cohort was included in the present study.

#### 2.1.2. Cross-Sectional Sample

To examine the group differences with regard to attention to face and its relation to clinical characteristics of our participants with ASD, we used a cross-sectional sample, in which we included the first timepoint assessment where the participants had a valid eye-tracking recording. The inclusion criteria for this cross-sectional sample were to have a confirmed diagnosis of ASD and to have a screen attendance of at least 50% of the stimulus presentation (total duration 28.9 s). Our cross-sectional sample was composed of 111 young children aged from 1.39 to 3.96 years old, comprising 95 children with ASD (2.81 ± 0.65 years old, 14 females) and 16 TD children (2.49 ± 0.82 years old, 4 females) (see Table 1 for a full description of the sample). It should be noted here that our cross-sectional sample was a convenience sample. For this reason, we did not conduct a power analysis a priori. However, following a reviewer’s comment, we ran a power analysis a posteriori for the cross-sectional sample using the software package G*Power 3 [33]. Given that the cross-sectional sample included more than 52 participants, there was adequate statistical power (>0.80) to detect large effects. However, in order to reach adequate power (>0.80) to detect moderate and small effect sizes, the sample size should be, respectively increased to 128 and 788 participants [34].

We also note that as the size of the TD sample was limited, it only served for initial between-group comparisons, and most of the analyses exploited the larger ASD group as our main interest was to explore their developmental change and treatment outcome with regard to attention to face at baseline. 

#### 2.1.3. Unstructured Longitudinal Sample

Then, we tested whether the level of social orienting was predictive of the developmental outcome. Out of the 95 children with ASD of our cross-sectional sample, we had longitudinal data available for 81 children, which included a total of 251 assessments (i.e., 81 children had at least two timepoints, 51 had at least 3 timepoints, 27 had at least 4 timepoints, and 11 had 5 timepoints). The timepoints are carried out at 6-month intervals. We then divided our longitudinal sample of 81 children according to their attention to face at baseline. We used the proportion of time spent looking at the actress’s face (% Face) with regard to the total video time. Using the median (*Mdn* = 48.06), we distinguished two subgroups: 41 participants with a lower level of attention to the face (*ASD-AF-*), and 40 participants with a higher level of attention to the face at baseline (*ASD-AF+*). 

As previously mentioned, we used a convenience sample of children who received a diagnosis before the age of 4.5 years old. All children enrolled in the present study received some type of intervention, varying in intensity, according to the local treatment availabilities and to their parents’ choice. By the design of our study, we collected the type and intensity of the intervention as well as the number of hours of intervention, without influencing its delivery. In our longitudinal subsample comprising 81 young children with ASD, 51 children received intensive 20h/w of individualized (one child one therapist) intervention following the ESDM program [29] (for a description of the intervention in this cohort, see [24]). Furthermore, 19 children out of 51 (37%) included in the *ASD-ESDM* group received other interventions in addition to 20 h of the ESDM per week (average of 21.6 h of treatment ranging from 21 to 24 h/w). Eight children had speech therapy (range 1–2 h/w), 8 children had occupational therapy (range 1–3 h/w), 2 children had both speech therapy and occupational therapy (duration 2 h/w), and 1 child had psychotherapy and speech therapy (duration 4 h/w).

Among the 81 children in the unstructured longitudinal sample, 30 received treatments available in the community (*ASD-CT*). Out of the 30 children in the *ASD-CT* group, 5 (17%) received speech therapy only (range 0.5–1.25 h a week), and 25 (83%) received multiple interventions, such as speech therapy and occupational therapy (range 1–14 h/w). Out of the 30 children, the average of the total intervention duration per week was 3.4 h and the range was between 0.5 and 14 h/w. 

In order to explore the effects of social orienting at baseline and of the type of intervention on treatment outcome, we subsequently divided the *ASD-AF+* and *ASD-AF-* groups into two subgroups according to the type of intervention. Children with a higher level of social orienting were distinguished according to early and intensive treatment (*ESDM-AF+*; N = 25; 2.83 ± 0.48 years old; 2 females) and community treatment (*CT-AF+*; N = 16; 3.04 ± 0.55 years old; 4 females). The same grouping was carried out for children with a lower level of social orienting receiving either early and intensive treatment (*ESDM-AF-*; N = 26; 2.68 ± 0.68 years old; 4 females) or community treatment (*CT-AF-*; N = 14; 2.62 ± 0.67 years old; 1 female). The 4 groups did not significantly differ by sex (*p* = 0.552) and age (*p* = 0.189).

### 2.2. Measures

The Autism Diagnostic Observation Schedule, 2nd version (ADOS-2) [32] is the gold-standard assessment for the diagnosis of ASD. The ADOS-2 standardizes the clinical observation of autism symptoms in three domains of behavior: communication, social interaction, and repetitive and restrictive behaviors or interests. Based on the participant’s age and language level, an appropriate ADOS-2 module was selected. Given that the modules of the ADOS-2 differed between participants, we used the ADOS-2 Calibrated Severity Score as a standardized measure of symptom severity across all modules [35,36]. Calibrated Severity Scores are also available for the two subdomains, namely Social Affect (SA Severity Score) and Restricted and Repetitive Behaviors (RRB Severity Score) [37]. The ADOS-2 were performed and coded by trained examiners. 

Further, to measure developmental functioning, we used the Mullen Scales of Early Learning [30] which is a standardized tool that can be administered to children from of 0 to 68 months. The scale includes five subdomains: Visual Reception (VR), Fine Motor Skills (FM), Gross Motor Skills (GM), Receptive Language (RL), and Expressive Language (EL). While the MSEL yields standardized T-scores for each domain, they can be of limited use for children with a neurodevelopmental disorder such as ASD [38]. Indeed, the T-scores showed a strong floor-level performance in the lower functioning end of our sample. For this reason, and following others [38], we used Age Equivalent (AE) scores as they demonstrate good variability and offer clear interpretation. Indeed, a developing child is expected to have an AE score similar to their own chronological age. In this study, we were interested in assessing verbal skills given the proximity of social and language development [39] and the impact of social orienting in both of these domains [40]. We used the EL AE and RL AE scores and we computed two global scores: (1) Total AE, which is the average of the four “cognitive” scales, RL, EL, FM, RV AE scores, and (2) Verbal AE score, obtained by averaging the RL and EL AE scores. 

Adaptive functioning was assessed using the Vineland Adaptive Behavior Schedule—2nd edition (VABS-II) [41], which is a semi-structured interview with the participants’ parents exploring adaptive functioning in the areas of communication, daily living skills, socialization, and motor skills. We computed an adaptive behavior composite (ABC) score of all these domains to obtain an indicator of overall adaptive functioning. Then, we looked more precisely into the scales of Communication, Socialization, and Daily Living Skills, for which we used the standardized scores. 

### 2.3. Procedure and Stimulus

Eye-tracking was conducted in a quiet room at the Autism Brain and Behavior Lab under constant light conditions. Participants were either seated on a chair or on their parents’ lap approximately 60 cm from the monitor. Gaze data were collected using Tobii Studio software 30.10.6 on a TX300 Tobii eye-tracker system with a 300 Hz sampling rate (Tobii Technology, Stockholm, Sweden). Each session began with a standard 5-point calibration procedure involving child-friendly animations. After the calibration phase, toddlers watched a 28.9 s video of an actress engaging in a child-directed speech. The stimulus covered the full size of a presentation screen with a height of 1200 pixels (29°38′) and width of 1920 pixels (45°53′). The video shows a close-up image of a woman speaking, engaging with simple sentences in French (e.g., “Hello, how are you doing? Are you fine? You look pretty today...”) along with joyful facial expressions (Figure 1a). The actress uses explicit cues for attention (e.g., nodding, asking the child questions, taking short breaks, asking for confirmation) that aim at attracting the child’s attention and responding to her initiating the social interaction. 

### 2.4. Analysis Strategy 

#### 2.4.1. Eye-Tracking Data Analysis

Tobii Studio was used to draw the area of interest (AOI) and to analyze gaze patterns. We drew one dynamic AOI over the actress’s face (Figure 1b). The actress stimulus had a height of 1023 pixels (25°25′) and a width of 786 pixels (19°40′) and the face AOI had a height of 325 pixels (8°11′) and a width of 244 pixels (5°43′). Accordingly, to examine the participants’ active attention to the face, we selected the measure of total fixation duration, defined as the sum of the duration for all fixations within an AOI. We selected the Tobii IV-T Fixation filter [42] to define fixations. The minimum fixation duration was 60 ms. We also controlled for overall time spent on the screen during stimulus presentation by calculating the percentage of time children fixated on the face AOI relative to overall time spent on the whole screen. We thus obtained dependent variables that were based on the proportions of total fixation duration of the face AOI that we denote “%Face”.

#### 2.4.2. Statistical Analyses

In order to explore how the attention to face at baseline was associated with clinical characteristics in terms of autistic symptoms and developmental and adaptive skills, we conducted Pearson correlation analyses between social orienting (measured by %Face) and clinical measures (ADOS-2, MSEL, VABS-II). Graphs were produced with GraphPad PRISM 80.0 (GraphPad Prism version 8.1.0 for Macintosh, GraphPad Software, San Diego, CA, USA).

To investigate whether levels of attention to face at baseline had an impact on developmental change over time, we used a mixed model method. Mixed modeling is an ideal method for nested data that include a variable number of timepoints [42]. We carried out a random slope mixed model analysis [43] implemented in MATLAB R2019b (Mathworks, Natick, MA, USA). First, we used a grouping variable of Attention to Face (AF). This grouping variable distinguished the participants according to their level of attention to face at baseline. To do this, we used the median (*Mdn* = 48.6) of the variable %Face, which denotes the proportions of total fixation duration of the face AOI. Based on this cutoff value, we differentiated children with a higher level of AF (i.e., higher than the median) at baseline (*ASD-AF+*), from children with a lower level of AF (i.e., lower than the median) at baseline (*ASD-AF-*). 

Then, we estimated differences in the developmental trajectories between the two subgroups (*ASD-AF+* and *ASD-AF-*) by fitting random slope models (constant, linear, or quadratic), which correspond to a different relationship between age and our four measures of development (which we denoted MSEL Total Age Equivalent, MSEL Verbal Age Equivalent, MSEL EL Age Equivalent, and MSEL RL Age Equivalent). Using the Bayesian information criterion allowed us to select the model with the best fit. For instance, we obtained a full linear model as follows: (1)$$Y_{ij}=\beta_{0}+u_{i}+\beta_{1}\times g_{i}+\beta_{2}\times x_{ij}+\beta_{3}\times u_{i}\times x_{ij}+\epsilon_{ij}$$*Y*: one of the four measures of development (MSEL Total Age Equivalent, MSEL Verbal Age Equivalent, MSEL EL Age Equivalent, and MSEL RL Age Equivalent). *i,j*: [subject, timepoint] index, β: fixed effects, *u*: normally distributed random effect, *x*: age, ϵ: normally distributed error term.

We evaluated the significance of the between-group differences in the intercept and in the slope by running a log-likelihood ratio test between the full model and any of the reduced models:Reduced group effect model:(2)$$Y_{ij}=\beta_{0}+u_{i}+\beta_{2}\times x_{ij}+\beta_{3}\times u_{i}\times x_{ij}+\epsilon_{ij}$$Reduced slope model:(3)$$Y_{ij}=\beta_{0}+u_{i}+\beta_{1}\times g_{ij}+\beta_{2}\times x_{ij}+\epsilon_{ij}$$

Consequently, we obtained a comparison between the intercept (i.e., group effect) and the slope of developmental trajectories (i.e., group × age interaction effect) of the developmental measure of each group. 

In order to explore whether children’s levels of attention to face at baseline had an impact on treatment outcome, whether the children were part of the ESDM group or the CT group, we conducted mixed model regression to identify the developmental trajectories for the four subgroups according to their levels of social orienting and the type of treatment received (*ESDM-AF+*; *ESDM-AF-*; *CT-AF+*; *CT-AF-*). Both mixed model analyses were computed by fitting random slope models to the data, taking into account both within-subject and between-subject effects.

## 3. Results

### 3.1. Group Differences at Baseline

The distribution of the attention to face was normal for the ASD group and the TD group (*p* > 0.05). The difference in the proportion of attention to the actress’s face (%Face) between the ASD (46.9 ± 22.2) and TD groups (54.9 ± 24.9) was not statistically significant (*t*_(109)_ = 1.319, *p* = 0.190).

### 3.2. Cross-Sectional Results

In the cross-sectional sample, we explored the relationship between social orienting of the ASD group and their baseline clinical characteristics in terms of the level of symptoms and developmental and adaptive skills.

#### Attention to Face and Clinical Characteristics

First, we examined the relationship between attention to face and autistic symptoms at baseline (Figure 2). Overall, less attention to the face was associated with a higher Total Severity Score (*r* = −0.292, *p* = 0.004, Figure 2a). This correlation was driven by the SA domain, as we identified a significant correlation between %Face and SA Severity Score (*r* = −0.364, *p* < 0.001, Figure 2b) and no significant correlation between %Face and RRB Severity Score (*r_s_* = 0.039, *p* > 0.05, Figure 2c). 

Second, regarding developmental skills at baseline, we found that children who proportionally spent more time on the face (%Face) showed better development (as indexed by the Total AE) across all domains (*r* = 340, *p* < 0.001, Figure 3a). More specifically, we identified significant correlations between the attention to the face and the Verbal AE scores (*r* = 0.360 *p* < 0.001, Figure 3b), and both the EL AE scores (*r* = 0.365, *p* < 0.001, Figure 3c) and the RL AE scores (*r* = 0.344, *p* < 0.00, Figure 3d). 

Third, based on the VABS-II, more time spent on the actress’s face (%Face) was significantly associated with better overall adaptive skills (ABC, *r* = 0.403, *p* < 0.001, Figure 4a). This correlation was mainly led by the domains of Daily Living Skills (*r* = 0.414, *p* < 0.001, Figure 4b) and Communication (*r* = 0.476, *p* < 0.001, Figure 4c), and to a lesser extent by the Socialization domain (*r* = 0.350 *p* < 0.001. Figure 4d). 

### 3.3. Longitudinal Analyses

#### 3.3.1. Attention to Face as a Predictor of Developmental Change over Time

To investigate the role of attention to face as a predictor of developmental gains, we conducted a mixed model regression. Using the MSEL as a measure of developmental skills, we found that children with more attention to face at baseline (*ASD-AF+*) demonstrated statistically significantly higher developmental scores over time. Overall, the *ASD-AF+* group showed higher scores across the Total AE score (*p_group_* = 0.010) than the *ASD-AF-* group (Figure 5a). We also found a significant interaction for Attention to face × Total AE (*p_interaction_* = 0.003), indicating that relative to the *ASD-AF-* group, the *ASD-AF+* group showed significantly greater developmental gains over time. When focusing on the verbal domains, we found that the *ASD-AF+* group had higher scores than the *ASD-AF-* group (*p_group_* = 0.022, see Figure 5b). Again, we found a significant interaction between attention to face × Verbal AE (*p_interaction_* = 0.006), suggesting that relative to the *ASD-AF-* group, the *ASD-AF+* group showed significantly greater language gains. Looking more closely, attention to face at baseline has more impact on expressive language skills (*p_group_* = 0.006, see Figure 5c) than on receptive language skills (*p_group_* = 0.108, see Figure 5d). Indeed, we found a significant interaction between attention to face × Expressive Language AE (*p_interaction_* = 0.002), indicating that the *ASD-AF+* group showed significantly greater gains in expressive language in comparison to the *ASD-AF-* group. 

To check whether this result was not driven by the baseline MSEL scores of the ASD-AF+ and ASD-AF- groups, we conducted another mixed model analysis by dividing children in the ASD group according to their MSEL performance at baseline (see Appendix A). We separated them into two groups using the Total AE baseline median (*Mdn* = 19). We found that children with a higher Total AE baseline score did not show significantly greater gains than the group of children with a lower Total AE score (*p*_interaction_ > 0.05, Figure A1). This result indicates that attention to face at baseline is a better predictor of development change than the MSEL performance at baseline. 

#### 3.3.2. Attention to Face as a Predictor of Treatment Outcome

To examine the role of attention to face as a predictor of treatment outcome, we conducted a mixed model regression with four groups, related to the level of baseline attention to face and type of treatment received (*ESDM-AF+*; *ESDM-AF-*; *CT-AF+*; *CT-AF*). We measured treatment outcome with the MSEL Age Equivalent scores.

Our results show faster developmental gains in the *ESDM-AF+* group in comparison to the three others (Figure 6), suggesting that higher levels of social orienting at baseline and early and intensive intervention are strong predictors for developmental gains over time. To check whether the significant results are driven by the *ESDM-AF+* group, we conducted post hoc mixed model analyses where we compared the four groups two by two (see Appendix B). When we did not take the *ESDM-AF+* group into account, the models were no longer significantly different (*p* > 0.05), indicating that this group showed significantly greater developmental gains over time (Figure A2).

Of note, in this study, we included in the longitudinal analyses children who were already included in an intervention program (i.e., the baseline eye-tracking measure did not correspond to the start of the intervention). Out of the 81 children included in the longitudinal sample, 29 had started treatment 6 to 18 months before the eye-tracking assessment (average time interval since intervention start: 9 months). Of these, eight children were from the *ESDM-AF+* group, nine children from the *ESDM-AF-* group, eight children from the *CT-AF+* group, and four children from the *CT-AF-* group. To verify that our results were not biased by the individuals who already had access to an intervention before the baseline eye-tracking measure, we carried out a supplementary analysis (see Appendix C) in which we only considered the remaining 52 children who had a baseline eye-tracking timepoint. We obtained the same structure of results with our mixed models with the four subgroups (Figure A3). We found a significant group effect and interaction for the measure of Total AE (*p_group_* < 0.001, *p_interaction_* < 0.001), Verbal AE (*p_group_* = 0.011, *p_interaction_* = 0.001), Expressive Language AE (*p_group_* = 0.025, *p_interaction_* = 0.003), and Receptive Language AE (*p_group_* = 0.020, *p_interaction_* = 0.002).

## 4. Discussion

In this study, we examined the relationship between social orienting and baseline clinical characteristics in young children with ASD, as well as explored the role of social orienting as a predictor of developmental change and treatment outcome. To do this, we measured social orienting in terms of attention to face with a 29 s video of an actress engaging in child-directed speech. First, our results indicate that attention to face is associated with the level of autistic symptoms, and with developmental and adaptive skills. Second, we found that a higher level of attention to face at baseline predicted developmental change in a longitudinal subsample. Third, our results show that social orienting at baseline modulated developmental change and treatment outcome. 

Our preliminary results regarding the cross-sectional ASD and TD samples corroborated our hypothesis, as we showed no statistically significant differences in terms of proportion of fixation time to the face AOI. Given the small TD sample size, it is important to interpret this result with caution. Nevertheless, it should be noted that some studies (e.g., [19,20]) did not observe differences in visual exploration in ASD children and their TD peers. Previous work [8,13,18,21,22,46] found that social context modulated visual exploration of children with ASD in comparison to their TD peers. Indeed, the social orienting impairment was more significant in children with ASD when the stimulus was more socially engaging (e.g., eye contact and child-directed speech) and when it had a higher social content (e.g., people interacting). This suggests that this impairment may be induced by difficulty in monitoring more people at once and social interactions. Another relevant interpretation of our result relies on facial emotion. One study [47] showed that children with ASD did not attend less to faces than their TD peers when they were presented with happy faces. Taken together, these results imply that differences in attention to face in children with ASD in comparison to their TD peers are subtle and context dependent [48].

As we hypothesized, we found that attention to face was negatively related to the autistic symptom severity, and positively associated with developmental and adaptive skills. Our correlation analyses showed that the children with ASD who fixated more on the face had fewer symptoms of autism. This correlation was led by symptoms related to the social communication and social interactions, and not by symptoms related to restricted and repetitive behaviors. In addition, children with higher levels of attention to face at baseline showed significantly better verbal skills in both expressive and receptive language, and they showed significantly better adaptive skills, notably in the areas of communication, socialization, and daily living skills, which is line with previous studies [6,8,41]. Importantly, our findings highlight that a very short and simple video can provide valuable information about a child’s behavioral phenotype and potentially about their future development as well. Indeed, the close relationship between attention to face and baseline clinical characteristics underlines that attention to face is a marker of difficulties in the social, verbal, and adaptive domains in early development, which is in line with the social motivation theory [9,10]. Deficits in social skills could produce detrimental long-term effects on the development of social cognition, which hinders participation in social interactions [49].

Our longitudinal results show that the eye-tracking measure of attention to face was a reliable predictor of developmental change. We found that, while the two subgroups (*ASD-AF+* and *ASD-AF-*) improved across all domains of development, children who had a higher level of attention to face (*ASD-AF+*) at baseline made the greatest progress. In line with our hypothesis and with previous studies [27,50], this finding indicates that attention to face at baseline is a predictor of developmental change across all domains. We also explored whether social orienting at baseline impacted verbal gains given the proximity of the social and language domains in early development [39,51]. We found that participants with ASD who oriented less toward the actress’s face at baseline (*ASD-AF-*) showed fewer gains, specifically in the verbal domain. In line with this finding, one longitudinal study [52] found that children with ASD with more socially unresponsive behaviors (e.g., not looking at faces, avoiding eye contact) at baseline made significantly less progress over two years in both vocabulary comprehension and production, and language comprehension. From this result, we can infer that the *ASD-AF-* group might have missed valuable social learning opportunities, which negatively impacted their social and language development over time [53]. Extending this result, we propose that children with less social orienting may not rely on social information to engage with the world [54] but, rather, they show a preference for the physical world. This orientation toward non-social information may promote the development of restricted interests, as has been recently reported in an neuroimaging study [55]. Consequently, this will likely create a vicious circle considering that engaging in rigid routines is likely to prevent acquisition through social learning [56]. 

Furthermore, we found that social orienting modulated the treatment outcome. Consistent with our hypothesis, our mixed model regression analyses indicated that among all children with ASD who received an ESDM intervention, those with higher levels of attention to face at baseline (*ESDM-AF*+) showed significantly greater developmental gains over time across all domains, and verbal domains specifically, in comparison to children with less attention to face (*ESDM-AF-*). Importantly, our findings are in line with previous results from our group [24] showing that children who spontaneously orient toward social stimuli would show an earlier response to the ESDM. Indeed, we suggest that the ESDM is more beneficial for children who spontaneously orient to social stimuli. Nevertheless, it is important to note that while a higher level of social orienting at the beginning of an intervention is a valuable asset, the ESDM intervention does aim to increase the social orienting of all children. We emphasize that enhancing orienting toward social stimuli will promote social learning, which will set off a positive developmental cascade. For this reason, we suggest that improving social orienting is an important first step of treatment as it influences the rate of developmental change [24]. 

While it is widely acknowledged that the ESDM is an effective intervention for young children with ASD [29], we propose that it leads to heterogeneity in outcome. This proposition aligns with Contaldo et al.’s study [57], which recently showed high variability in the outcomes of children with ASD after receiving one year of an ESDM intervention. To better understand this heterogeneity, previous studies [56,57] have reported the role of several predictors of treatment outcomes (e.g., non-verbal abilities, symptom severity, and goal understanding). However, studies either did not include attention to faces in their analysis [57], or their results regarding its role as a predictor were not significant [56]. In this context, our study adds a new perspective to understand heterogeneity in treatment outcome, since we suggest that a simple behavioral measure is a valuable predictor leading to heterogeneous developmental trajectories. Contrasting with our finding, Vivanti et al. [56]’s eye-tracking study found that attention to faces did not predict treatment response in young children with ASD. According to the authors, a possible explanation for this result relies on the nature of the eye-tracking task. While their task consisted of an actor moving his hand toward background objects, our task featured an empty background with an actress speaking in child-directed speech. Thus, we argue that our stimulus being more engaging is of greater relevance to measure children’s spontaneous social attention. Indeed, the actress’s intense, positive expressions of emotion in the video might have helped in gaining the children’s attention. We previously showed that an intense emotion of surprise helped children with ASD to better follow gaze shifts in an eye-tracking task [58]. Taken together, our findings indicate that intense facial expressions are important to socially engage children with ASD, as social attention promotes social learning experiences [9,10]. The intensity of emotions is a particularly essential tool in the ESDM intervention, given that the core principle of the model is to socially engage the child through positive affect and interpersonal exchange [29]. By socially engaging children, the ESDM intervention aims to increase the salience of social stimuli, as children with ASD may have failed to assign them a reward value, in order to enhance their social motivation to initiate and respond to social interaction [59].

In the framework of precision medicine, we suggest that a first step toward an individualized early and intensive intervention is to identify early predictors of treatment outcome [56]. Building on this framework, our study provides preliminary data suggesting the relevance of a simple behavioral measure of attention to face as a valuable predictor of treatment outcome, as it leads to heterogeneous developmental trajectories. 

Our study has several limitations. First, one of them relates to the fact that we used a convenience sample, so systematic sample bias in treatment type cannot be entirely excluded. This is particularly true for the CT group, who had access to highly heterogeneous types of treatment. However, a large part of our sample received a homogenous and intensive type of treatment, and our results provide strong support for the fact that social orienting modulates response to this type of treatment. Second, power calculations demonstrated that our sample size was sufficient to detect large effect sizes, but not moderate and small effect sizes, so that our study might have left other effects undetected. Future studies with larger longitudinal samples are critical to better delineate predictors of treatment response for preschoolers with autism. Third, as the small size of our TD sample prevents us from generalizing results to the TD population, our result regarding group differences should be considered with caution. Fourth, our ASD sample includes more males than females. This is indeed a common limitation of many studies on autism, given the sex ratio imbalance in autism [60]. To verify that the inclusion of females in our sample did not affect the result, we repeated all the analyses (cross-sectional and longitudinal) after excluding the females. As shown in Appendix D, all the results remained significant when including the males only (Figure A4, Figure A5, Figure A6, Figure A7 and Figure A8). Future studies including a larger female sample are required to verify whether the effect shown in our studies is only specific for males with ASD, or holds in samples of females with ASD as well.

## 5. Conclusions

Our study examined the relationship between social orienting and clinical characteristics at baseline, and the role of social orienting to predict developmental change and treatment outcome. On the cross-sectional level, we found that attention to face is robustly correlated with autistic symptoms and developmental and adaptive skills. Moreover, our longitudinal results suggest that attention to face is a predictor of developmental change and verbal gains in particular. Most importantly, our results underline the importance of exploring social orienting in parallel with the type of treatment received. Our study suggests that social orienting predicts a better treatment outcome in the context of an early and intensive intervention, paving the way toward individualized interventions and precision medicine.

## Figures and Tables

**Figure 1 biomedicines-09-00942-f001:**
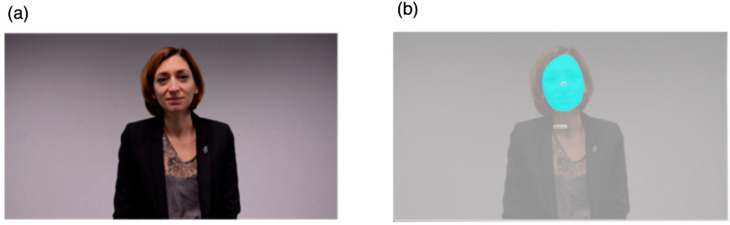
(**a**) Eye-tracking stimuli and (**b**) area of interest.

**Figure 2 biomedicines-09-00942-f002:**
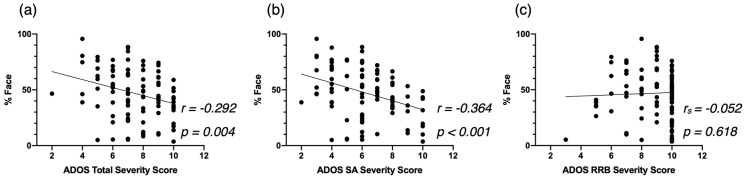
Correlation between %Face and the Autism Diagnostic Observation Schedule—2nd (ADOS-2) Total Severity Score (**a**), the ADOS-2 SA Severity Score (**b**), and the ADOS-2 RRB Severity Score (**c**), at baseline for the cross-sectional Autism Spectrum Disorder (ASD) group.

**Figure 3 biomedicines-09-00942-f003:**
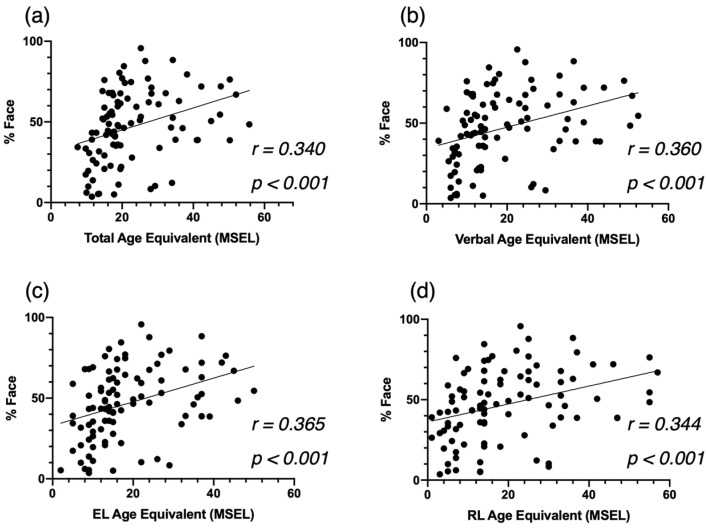
Correlation between %Face and the Mullen Scales of Eearly Learning (MSEL) Total AE (**a**), Verbal AE (**b**), Expressive Language AE (**c**), and Receptive Language AE (**d**), at baseline for the cross-sectional ASD group.

**Figure 4 biomedicines-09-00942-f004:**
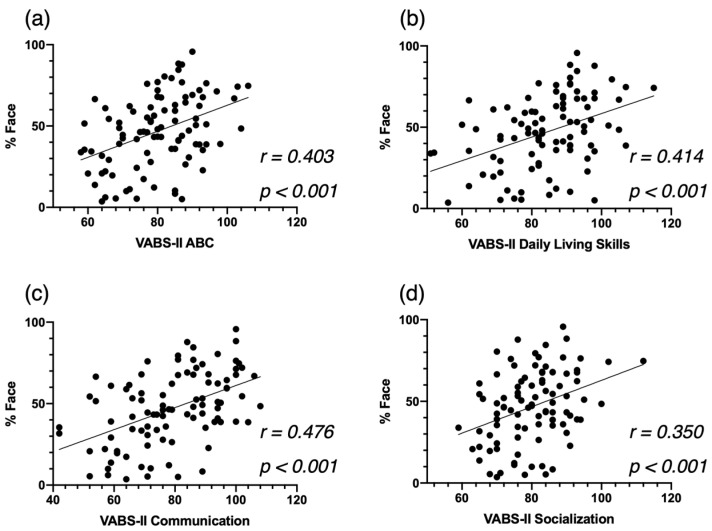
Correlation between %Face and the Vineland Adaptive Behavior Schedule—2nd edition (VABS-II) Adaptive Behavior Composite Score (**a**), the Communication Standardized Score (**b**) and the Socialization Standardized Score (**c**), at baseline for the cross-sectional ASD group, and the Daily Living Skills Standardized Score (**d**).

**Figure 5 biomedicines-09-00942-f005:**
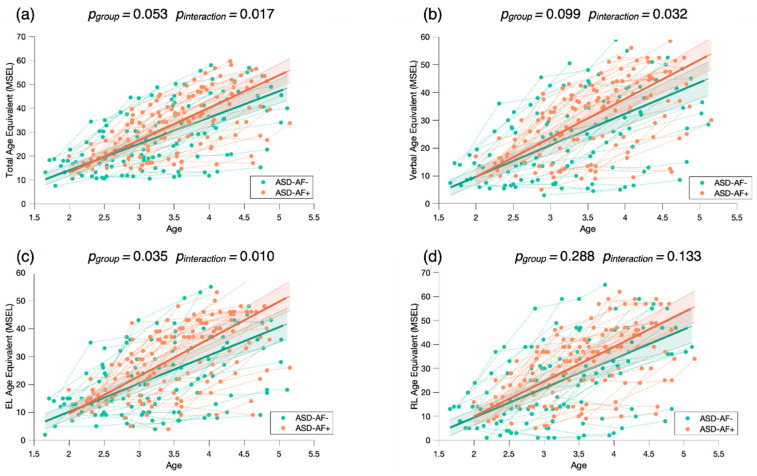
Mixed model regression analysis of the MSEL Total Age Equivalent (**a**), Verbal Age Equivalent (**b**), Expressive Language Age Equivalent (**c**), and Receptive Language Age Equivalent (**d**) for 265 longitudinal recordings from 81 young children with ASD (2.78 ± 0.65 years old) who demonstrated a lower (*ASD-AF-*) or higher (*ASD-AF+*) attention to face (AF) at baseline. The group-level trajectory (bold line) is derived from mixed-effect modeling [43,44,45]. The 95% confidence interval is indicated by the colored bands around the estimated group-level trajectory.

**Figure 6 biomedicines-09-00942-f006:**
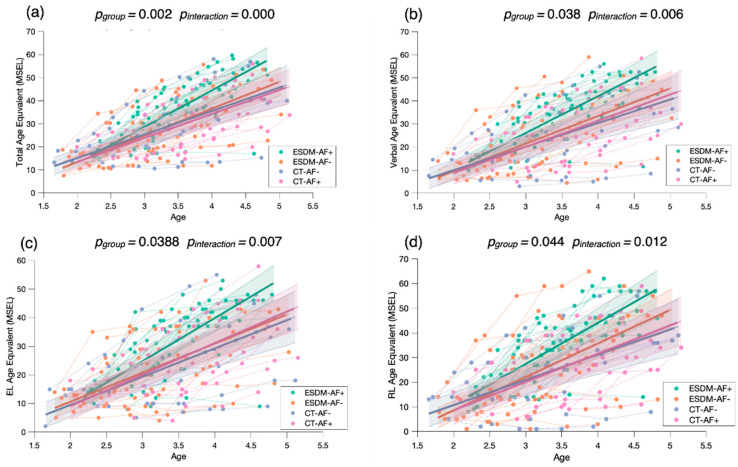
Mixed model regression analysis for the MSEL Total Age Equivalent (**a**), Verbal Age Equivalent (**b**), Expressive Language Age Equivalent (**c**), and Receptive Language Age Equivalent (**d**) for 265 longitudinal recordings from 81 young children with ASD (2.78 ± 0.65 years old) with lower or higher social orienting at baseline and who either received an early and intensive intervention (*ESDM-AF+; ESDM-AF-*) or community treatment (*CT-AF*+; *CT-AF-*). The group-level trajectory (bold line) is derived from mixed-effect modeling [43,44,45]. The 95% confidence interval is indicated by the colored bands around the estimated group-level trajectory.

**Table 1 biomedicines-09-00942-t001:** Cross-sectional and longitudinal sample demographics at baseline.

	Cross-Sectional Sample			Longitudinal Sample				
	ASD	TD	*p*	ESDM *AF+* ^4^	ESDM *AF-* ^5^	CT *AF+* ^6^	CT *AF-* ^7^	*p*
N	95	16		25	26	16	14	
Sex	14 F	4 F	0.290	2 F	4 F	4 F	1 F	0.552
Age (M ± SD)	2.81 ± 0.65	2.49 ± 0.82	0.096	2.83 ± 0.48	2.68 ± 0.68	3.04 ± 0.55	2.62 ± 0.67	0.189
ADOS-2 ^1^ Total	7.42 ± 1.81	1.25 ± 0.58	<0.001	6.92 ± 1.71	7.88 ± 1.70	7.25 ± 1.73	7.64 ± 2.24	0.273
ADOS-2 SA	6.33 ± 2.02	1.25 ± 0.58	<0.001	5.80 ± 1.71	7.08 ± 1.98	5.56 ± 1.93	6.50 ± 2.07	0.042
ADOS-2 RRB	8.74 ± 1.63	2.19 ± 2.17	<0.001	8.80 ± 1.29	8.31 ± 1.81	9.38 ± 1.09	8.71 ± 2.30	0.248
MSEL ^2^ Total AE ^3^	22.18 ± 11	35.4 ± 13	<0.001	24.1 ± 9.87	19.9 ± 9.50	24 ± 9.13	21.1 ± 11.1	0.384
MSEL Verbal AE	19.1 ± 12.2	33.9 ± 14.1	<0.001	21 ± 10.9	16.2 ± 10.3	20.2 ± 11.1	15 ± 10.7	0.230

^1^ ADOS-2, Autism Diagnostic Observation Schedule, 2nd edition, ^2^ MSEL, Mullen Scales of Early Learning, ^3^ AE, Age Equivalent score, ^4^ ESDM-AF+, children with more attention to face at baseline receiving early and intensive intervention, ^5^ ESDM-AF-, children with less attention to face at baseline receiving early and intensive intervention, ^6^ CT-AF+, children with more attention to face at baseline receiving community treatment, ^7^ CT-AF-, children with less attention to face at baseline receiving community treatment.

## Data Availability

Given that the collected data contain sensitive information of the families involved in the study, the data are not available publicly to avoid breach of patient confidentiality.

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
