# Peer review of "Attention to Face as a Predictor of Developmental Change and Treatment Outcome in Young Children with Autism Spectrum Disorder"

_biomedicines, 2021, doi:10.3390/biomedicines9080942_

Round 1

Reviewer 1 Report

Dear Authors,

Thanks a lot for the opportunity you have offered me to revise the manuscript “Attention to Face as a Predictor of Developmental Change and Treatment Outcome in Young Children with Autism Spectrum Disorder”. I thank the authors for their effort in producing this research during the COVID-19 pandemic.

Please find below constructive comments that may help to improve the clarity of the manuscript and the scientific soundness. 

# Introduction:

The introduction is well presented.

In the first paragraph, please consider that another early sign of  ASD in children may be the sensorimotor organization of  posture, e.g., Perin C, Valagussa G, Mazzucchelli M, et al. Physiological Profile Assessment of Posture in Children and Adolescents with Autism Spectrum Disorder and Typically Developing Peers. Brain Sci. 2020;10(10):681. Published 2020 Sep 27. doi:10.3390/brainsci1010068

# Methods:

I understand the difficulties for sample recruitment, especially for children with Autism Spectrum Disorders.  However, it is really difficult to follow how the sample was enrolled. It seems that the present study is a nested cross-section study within a cohort longitudinal study.

Consider to divide the 2.1. Participants section in more than on sub-heading. Now, it is  difficult to follow. A flow chart may help.

Please clarify: 1) the study design, 2) were the subjects evaluated multiple times? 3) Please describe the treatment that the sub-sample of the longitudinal study was subjected.

Since in the Introduction are presented three AIMS. I suggest to link the objectives in the study design and when explaining the Statistical analysis

# # 2.4.2. Statistical analyses

Clarify if a power analysis, i.e., a sample size computation, was run to account for the minimum number of subjects to enroll.  Detail the power and how the sample size was calculated.

Were the requirements for the slope mixed model analysis and the mixed model regression met?  What was the hierarchical structure for the model?

Detail the dependent / predictor and independent variables for all the analyses

An explanation about how to interpret the Statistical analyses is warranted.

Were the children subjected to multiple treatment adjusted in the analysis?  

# Results:

All the supplementary analyses may be presented under a specific sub-heading, this may help the reader to follow the text

# Discussion and Conclusions:

No comments.

Reviewer 2 Report

I would like to thank the authors for giving me the opportunity to review their fine manuscript. I enjoyed reading this and found it overall well done. 

My only real comment/suggestion has to do with the Introduction. In the last few lines of the Introduction on page 3, the sentence starting with "Based on findings from previous studies..." appears to have a type/information missing. I would request the authors clarify these hypotheses. Otherwise the manuscript appears to be well written and very thorough in its research approach and analyses. 
